# DeformUX-Net: Exploring a 3D Foundation Backbone for Medical Image Segmentation with Depthwise Deformable Convolution

## Abstract

The application of 3D ViTs to medical image segmentation has seen remarkable strides, somewhat overshadowing the budding advancements in Convolutional Neural Network (CNN)-based models. Large kernel depthwise convolution has emerged as a promising technique, showcasing capabilities akin to hierarchical transformers and facilitating an expansive effective receptive field (ERF) vital for dense predictions. Despite this, existing core operators, ranging from global-local attention to large kernel convolution, exhibit inherent trade-offs and limitations (e.g., global-local range trade-off, aggregating attentional features). We hypothesize that deformable convolution can be an exploratory alternative to combine all advantages from the previous operators, providing long-range dependency, adaptive spatial aggregation and computational efficiency as a foundation backbone. In this work, we introduce 3D DeformUX-Net, a pioneering volumetric CNN model that adeptly navigates the shortcomings traditionally associated with ViTs and large kernel convolution. Specifically, we revisit volumetric deformable convolution in depth-wise setting to adapt long-range dependency with computational efficiency. Inspired by the concepts of structural re-parameterization for convolution kernel weights, we further generate the deformable tri-planar offsets by adapting a parallel branch (starting from $1 \times 1 \times 1$ convolution), providing adaptive spatial aggregation across all channels. Our empirical evaluations reveal that the 3D DeformUX-Net consistently outperforms existing state-of-the-art ViTs and large kernel convolution models across four challenging public datasets, spanning various scales from organs (KiTS: 0.680 to 0.720, MSD Pancreas: 0.676 to 0.717, AMOS: 0.871 to 0.902) to vessels (e.g., MSD hepatic vessels: 0.635 to 0.671) in mean Dice. The source code with our pre-trained model is available at ****.

## 1 Introduction

Recent advancements have seen the integration of Vision Transformers (ViTs) Dosovitskiy et al. (2020) into 3D medical applications, notably in volumetric segmentation benchmarks Wang et al. (2021); Hatamizadeh et al. (2022b); Zhou et al. (2021); Xie et al. (2021). What makes ViTs unique is their absence of image-specific inductive biases and their use of multi-head self-attention mechanisms. The profound impact of ViTs has somewhat eclipsed the emerging techniques in traditional Convolutional Neural Network (CNN) architectures. Despite the limelight on 3D ViTs, large kernel depthwise convolution presents itself as an alternative for extracting features with a broad field of view and scalability Lee et al. (2022). Unlike standard convolutions, depthwise convolution operates on each input channel separately, leading to fewer parameters and enhancing the feasibility of using large kernel sizes. In comparing CNNs to ViTs, we observe that a key attribute for generating fine-grained dense predictions is extracting meaningful context with a large effective receptive field (ERF). However, beyond leveraging large ERF, core operators like global-local self-attention mechanisms and large kernel convolution each have their own sets of trade-offs. Local self-attention in hierarchical transformers struggles to provide long-range dependencies, while global attention in Vanilla ViTs exhibits quadratic computational complexity relative to the image resolution. Concurrently, large kernel convolution computes features using static summations and falls short in providing spatial aggregation akin to the self-attention mechanism. Given these observations, we

can summarize these trade-offs specifically for volumetric segmentation in three main directions: **1) global-local range dependency**, **2) adaptive spatial aggregation across kernel elements**, and **3) computation efficiency**. We further pose a question: **"Can we design a core operator that addresses such trade-offs in both ViTs and large kernel convolution for 3D volumetric segmentation?"**

Recent advancements, such as those by Ying et al. and Wang et al. Ying et al. (2020); Wang et al. (2023), have enhanced deformable convolution design, offering a computationally scalable approach that marries the benefits of Vision Transformers (ViTs) and CNNs for visual recognition tasks. Drawing inspiration from these advancements, we revisit deformable convolution to explore its capability in: (1) **efficiently adapting long-range dependencies and offering adaptive spatial aggregation with 3D convolution modules**, (2) **achieving state-of-the-art (SOTA) performance across diverse clinical scenarios, including organs, tumors, and vessels**, and (3) **setting a novel direction in the development of foundational convolution backbones for volumetric medical image segmentation.** Diverging from the likes of SwinUNETR Hatamizadeh et al. (2022a) and 3D UX-Net Lee et al. (2022), we introduce a pioneering architecture, termed 3D DeformUX-Net. This model aims to address challenges from convolution to global-local self-attention mechanisms and bolster the implementation of deformable convolution for 3D segmentation, ensuring robust performance in variable clinical sceanrios. Specifically, we leverage volumetric deformable convolution in a depth-wise setting to adapt long-range dependency handling with computational efficiency. Preceding the deformable convolution operations, we incorporate concepts from structural re-parameterization of large kernel convolution weights Ding et al. (2022) and present a parallel branch design to compute the deformable tri-planar offsets, ensuring adaptive spatial aggregation for deformable convolution across all feature channels. We evaluate 3D DeformUX-Net on supervised volumetric segmentation tasks across various scales using four prominent datasets: 1) MICCAI 2019 KiTS Challenge dataset (kidney, tumor and cyst) Heller et al. (2019), 2) MICCAI Challenge 2022 AMOS (multi-organ) Ji et al. (2022), 3) Medical Segmentation Decathlon (MSD) Pancreas dataset (pancreas, tumor) and 4) hepatic vessels dataset (hepatic vessel and tumor) Antonelli et al. (2022). 3D DeformUX-Net consistently outperforms current transformer and CNN SOTA approaches across all datasets, regardless of organ scale. Our primary contributions can be summarized as follows:

- We introduce the 3D DeformUX-Net, a pioneering approach that addresses the trade-offs inherent in the global-local self-attention mechanisms and convolutions for volumetric dense predictions. To the best of our knowledge, this represents the inaugural block design harnessing 3D deformable convolution in depth-wise setting, rivaling the performance of established transformer and CNN state-of-the-art (SOTA) models.

- We leverage deformable convolution in depth-wise setting with tri-planar offsets computed in a parallel branch design to adapt long-range dependency and adaptive spatial aggregation with efficiency. To our best knowledge, this is the first study to introduce multi-planar offsets into deformable convolution for medical image segmentation.

- We use four challenging public datasets to evaluate 3D DeformUX-Net in direct training scenario with volumetric multi-organ/tissues segmentation across scales. 3D DeformUX-Net achieves consistently improvement across all CNNs and transformers SOTA.

## 2 TIMELINE FOR SEGMENTATION NETWORK: FROM CNNS TO VITS

In the realm of medical image segmentation, 2D/3D U-Net is the starting point to demonstrate the feasibility of performing dense prediction with supervised training Ronneberger et al. (2015); Çiçek et al. (2016). A variety of network structure designs also propose (e.g., V-Net Milletari et al. (2016), UNet++ Zhou et al. (2018), H-DenseUNet Li et al. (2018), SegResNet Myronenko (2018)) to adapt different imaging modalities and organ semantics. Moreover, nnUNet is proposed to provide a complete hierarchical framework design to maximize the coarse-to-fine capabilities of 3D UNet Isensee et al. (2021). However, most of the networks only leverage the convolution mechanism with small kernel sizes and limit to the learning of locality with small ERF. Starting from 2021, the introduction of ViTs provides the distinctive advantages of long-range dependecy (global attention) and large ERF for different medical downstream tasks. There's been a substantial shift towards incorporating ViTs for enhanced dense predictions (e.g., TransUNet Chen et al. (2021), LeViT Xu et al. (2021), CoTr Xie et al. (2021), UNETR Hatamizadeh et al. (2022b)). Due to the quadratic complexity of

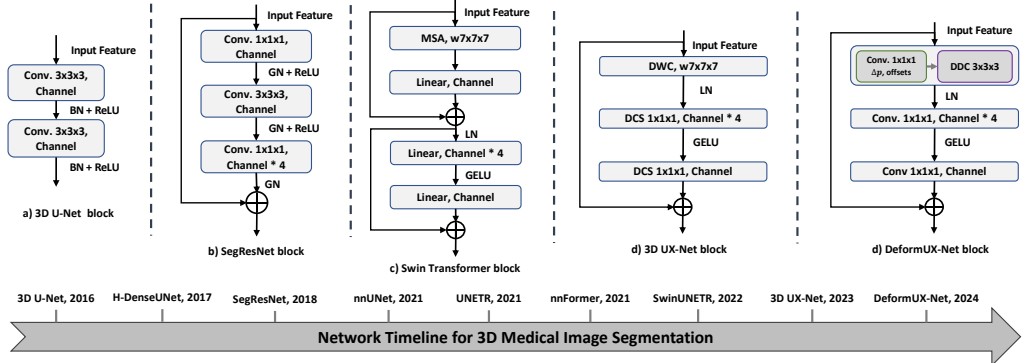

Figure 1: This figure compares our proposed block design with representative 3D medical image segmentation designs. We leverage depth-wise deformable convolution in parallel with a multi-layer perceptron (MLP), which generates the tri-planar offset to adapt long-range dependency and adaptive spatial aggregation across for deformable convolution. Furthermore, the deformable convolution module is followed with a MLP to provide linear scaling similarly to the Swin Transformer module.

multi-head self-attention mechanism in ViTs, it is challenging to adapt ViT as the foundation backbone for medical image segmentation with respect to high-resolution medical images. To tackle the computation complexity in ViTs, the hierarchical transformers (e.g., swin transformer Liu et al. (2021)) demonstrate notable contributions to extract fine-grain features with the concept of sliding window. Works like SwinUNETR Hatamizadeh et al. (2022a) and nnFormer Zhou et al. (2021), directly employ Swin Transformer blocks within the encoder to improve organ and tumor segmentation in 3D medical images. Some advancements, like Tang et al.'s self-supervised learning strategy for SwinUNETR, highlight the adaptability of these frameworks Tang et al. (2022). Similarly, 2D architectures like Swin-Unet Cao et al. (2021) and SwinBTS Jiang et al. (2022) incorporate the Swin Transformer to learn intricate semantic features from images. Yet, despite their potential, these transformer-based volumetric segmentation frameworks are bogged down by lengthy training times and considerable computational complexity, especially when extracting multi-scale features. In parallel with hierarchical transformers, depthwise convolution demonstrates an alternative to adapt large kernel sizes with efficiency. Notably, ConvNeXt by Liu et al. offers a glimpse of how one can blend the advantages of ViTs with large kernel depthwise convolution for downstream visual recognition tasks Liu et al. (2022). 3D UX-Net further bridges the gap to adapt large kernel depthwise convolution for volumetric segmentation with high-resolution medical images Lee et al. (2022). However, such large kernel design is limited to address the clinical scenarios of segmenting multi-scale tissues (e.g., tumors, vessels) Kuang et al. (2023) and additional prior knowledge may need to enhance learning convergence for locality in the large kernel Lee et al. (2023). Yet, a gap still persists in the literature regarding the feasibility and the optimal approach to adapt deformable convolution in volumetric segmentation. Given the advantages offered by the deformable convolution, there is potential to tackle most of the trade-offs across ViTs and CNNs with specific design.

## 3 3D DEFORMUX-NET: INTUITION

Inspired by Ying et al. (2020) and Wang et al. (2023), we introduce 3D DeformUX-Net, a purely volumetric CNN that revisits the concepts of deformable convolution and preserves all advantages across ViTs and CNNs mechanisms. We explore the fine-grained difference between convolution and the self-attention from local-to-global scale and investigate the variability of block design in parallel as our main intuition in Figure 1. With such observation, we further innovate a simple block design to enhance the feasibility of adapting 3D deformable convolution with robustness for volumetric segmentation. First, we investigate the variability across the properties of self-attention and convolution as three folds:

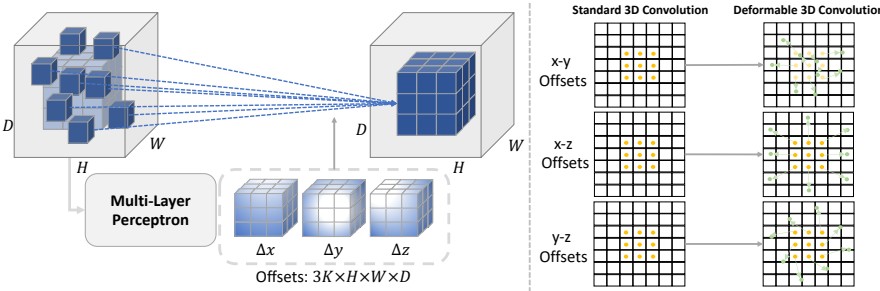

Figure 2: Overview of the deformable convolution mechanisms. Deformable convolutions introduce an adaptable spatial sampling capability that transcends the rigid bounds of conventional $3 \times 3 \times 3$ regions, achieved with the deformable offsets (light green arrows). Such offsets can demonstrate the capability of generalizing various transformation such as scaling and rotation (as shown in x-z, y-z offsets grid). The deformable offsets in our scenario are computed with a multi-layer perceptron.

- **Global-to-Local Range Dependency**: The concept of long-range dependency can be defined as more portion of the image is recognized by the model with a large receptive field. In medical domain, the de-facto effective receptive field for traditional segmentation network (e.g., 3D U-Net) is relatively small with the convolution kernel sizes of $3 \times 3 \times 3$. With the introduction of ViTs, the idea of transforming a $16 \times 16 \times 16$ patch into a 1-D vector significantly enhances the ERF for feature computation and defines it as global attention. While such global attention is limited to demonstrate the fine-grained ability for dense prediction, hierarchical transformer (e.g., Swin Transformer) is further proposed to compute local attention with large sliding window sizes specific for high-resolution segmentation. Meanwhile, large kernel convolution starts to explore in parallel, which shows the similar ability of hierarchical transformer and demonstrates the effectiveness of enlarging ERF in downstream segmentation tasks. However, the segmentation performance becomes saturated or even degraded when scaling up the kernel sizes. Therefore, an optimal ERF is always variable depending on the morphology of the semantic target for downstream segmentation.

- **Adaptive Spatial Aggregation**: While the weights computed from self-attention mechanism are dynamically conditioned by the input, a static operator is generated from the convolution mechanism with high inductive biases. Such characteristic enhances the recognition of locality and neighboring structure and leverages fewer training samples compared to ViTs. However, summarizing the visual content into a static value is limited to providing element-wise importance in a convolution kernel. Therefore, we hypothesize that an additional variant of prior information can be extracted within the visual context and may benefit the element-wise correspondence of the kernel weights.

- **Computation Efficiency**: Compared to both ViTs and CNNs, global attention from the traditional vanilla ViT demonstrates the lowest computation efficiency and it is challenging to scale up with respect to the quadratic complexity from the input size. Although the hierarchical transformers further reduce the feature dimensionality with sub-window shifting to compute self-attention, the computation of shifted window self-attention is computational unscalable to achieve via traditional 3D model architectures. Therefore, the depthwise convolution with large kernel sizes demonstrates to be another efficient alternative for computing features.

## 4 3D DEFORMUX-NET: COMPLETE BACKBONE

### 4.1 DEPTHWISE DEFORMABLE CONVOLUTION WITH TRI-PLANAR OFFSETS

To accommodate all trade-offs that we explored from both convolution and self-attention mechanisms, the simplest way is to innovate a block design that can bridge the gap, adapting long-range

dependency and adaptive spatial aggregation with efficiency. We hypothesize that a variant of convolution, deformable convolution, can provide the feasibility to address all trade-offs. Given a volumetric input $x \in \mathcal{R}^{C \times H \times W \times D}$ and the current centered voxel $v_0$ in the kernel, the operation of deformable convolution can be formulated as:

$$y(v_0) = \sum_{k=1}^{K} w(v_k) \cdot x(v_0 + v_k) \tag{1}$$

where $v_0$ represents an arbitrary location in the output feature $y$ and $v_k$ represents the $k_{th}$ value in the convolution sampling grid $G = (-1, -1, -1), (-1, -1, 0), ..., (1, 1, 0), (1, 1, 1)$ with $3 \times 3 \times 3$ convolution kernel as the foundation basis. $K = 27$ is the size of the sampling grid. $w \in \mathcal{R}^{C \times C}$ denotes the projection weight of the k-th sampling point in the convolution kernel. To enlarge the receptive field of a $3 \times 3 \times 3$ convolution, adapting learnable offsets is the distinctive properties for deformable convolution and enhance the spatial correspondence between neighboring voxels, as shown in Figure 2. Unlike the videos input in Ying et al. (2020), 3D medical images provide high-resolution tri-planar spatial context with substantial variability organs/tissues morphology across patients' conditions. To adapt such variability, we propose to adapt tri-planar learnable offsets $\Delta v_k \in \mathcal{R}^{3K \times H \times W \times D}$, which has a channel size of $3K$ and each $K$ channel represent one of axes (i.e., height, width and depth) for 3D spatial deformation. Furthermore, we observe that the offset computation mechanism have similar block design to the parallel branch design for structural re-parameterization to adapt large kernel convolutions Ding et al. (2022). With such inspiration, instead of using standard deformable convolution for both feature and offset computation, we propose to adapt deformable convolution in depthwise setting to simulate the self-attention behavior. Furthermore, we adapt parallel branch design to re-parameterize the offsets computation with either a Multi-Layer Perceptron (MLP) or small kernel convolution, enhancing the adaptive spatial aggregation with efficiency. We define our proposed deformable convolution mechanism at layer $l$ as follows:

$$\Delta v_0 = MLP(y^{l-1}(v_0))$$
$$y^l(v_0) = \sum_{g=1}^{G} \sum_{k=1}^{K} w_g(v_k) \cdot x_g(v_0 + v_k + \Delta v_0) \tag{2}$$

where $G$ defines as the total number of aggregation groups in the depth-wise setting. For the g-th group, $w_g \in \mathcal{R}^{C \times C'}$ denotes as the group-wise divided projection weight. Such deformable convolution operator demonstrates the merits as follows: 1) it tackles the limitation of standard convolution with respect to long-range dependencies and adaptive spatial aggregation; 2) it inherits the inductive bias characteristics of the convolution mechanism with better computational efficiency using less training samples.

## 4.2 Complete Architecture

To benchmark the effectiveness of each operation module fairly, inspired by 3D UX-Net, we follow the step-by-step comparison to compare the effectiveness of each module and create the optimal design with our proposed deformable operator. Given random patches $p_i \in \mathcal{R}^{H \times W \times D \times C}$ are extracted from each 3D medical images $x$, we follow the similar architecture with the encoder in 3D UX-Net, which first leverage a large kernel convolution layer to compute the partitioned feature map with dimension $\frac{H}{2} \times \frac{W}{2} \times \frac{D}{2}$ and project to a $C = 48$-dimensional space. We then directly substitute the main operator of large kernel convolution with our proposed Depth-wise Deformable Convolution (DDC) using kernel size of $3 \times 3 \times 3$. Furthermore, we further perform experiments to evaluate the adeptness of linear scaling in standard (MLP) and depth-wise setting (scaling approach in 3D UX-Net), considering MLP as the main scaling operator for DDC based on the performance evaluation. We hypothesize such block design can tackle all trade-offs across ViTs and large kernel convolution with robust performance and large ERF. Here, we define the output of the encoder blocks in layers $l$ and $l + 1$ as follows:

$$\hat{z}^l = \text{DDC}(\text{LN}(z^{l-1})) + z^{l-1}$$
$$z^l = \text{MLP}(\text{LN}(\hat{z}^l)) + \hat{z}^l$$
$$\hat{z}^{l+1} = \text{DDC}(\text{LN}(z^l)) + z^l$$
$$z^{l+1} = \text{MLP}(\text{LN}(\hat{z}^{l+1})) + \hat{z}^{l+1} \tag{3}$$

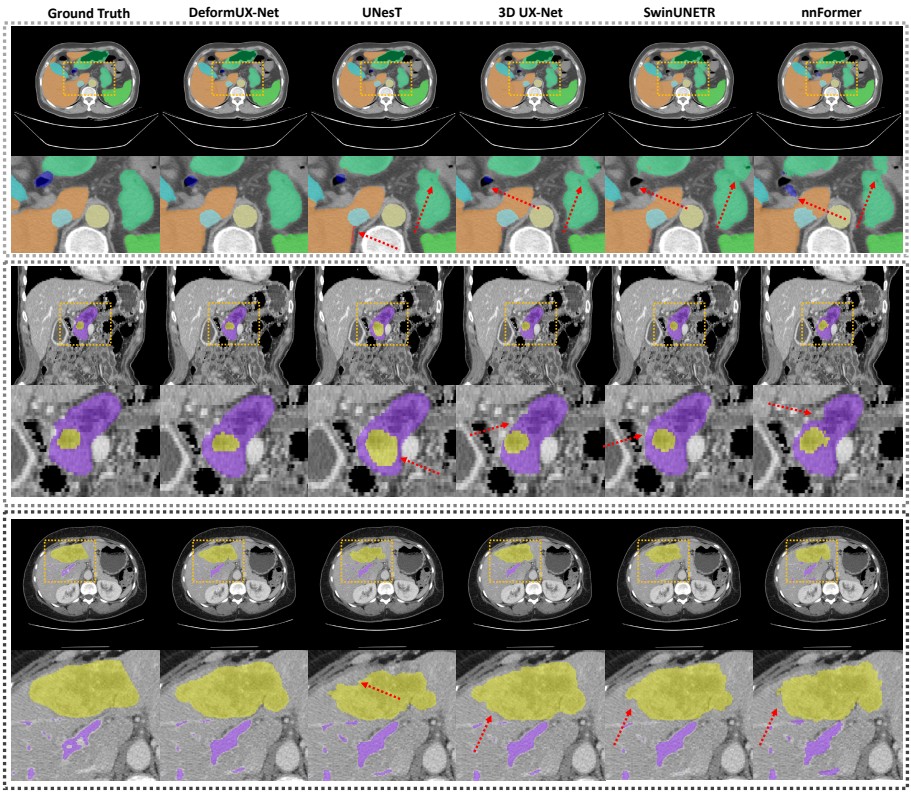

Figure 3: Qualitative representations are showcased across the AMOS, MSD pancreas, and hepatic vessels datasets. Selected areas are magnified to highlight the notable discrepancies in segmentation quality, with red arrows indicating areas of over-/under-segmentation. Overall, DeformUX-Net demonstrates the best segmentation quality compared to the ground-truth.

where $\hat{z}_l$ and $\hat{z}_{l+1}$ are the outputs from the DDC layer in different depth levels; LN denotes as the layer normalization. Compared to the 3D UX-Net, we substitute the large kernel convolution modules with two DDC layers. More details of the remaining architecture are provided in the supplementary material.

## 5 EXPERIMENTAL SETUP

**Datasets** We conduct experiments on five public segmentation datasets across organs/tissues from different scales (large (e.g., liver, stomach) to small (e.g., tumors, vessels), which comprising with 1) MICCAI 2019 KiTS Challenge dataset (KiTS) Heller et al. (2019), 2) MICCAI 2022 AMOS Challenge dataset (AMOS) Ji et al. (2022), 3) Medical Segmentation Decathlon (MSD) pancreas dataset and 4) hepatic vessel dataset Antonelli et al. (2022). For KiTS dataset, we employ 210 contrast-enhanced abdominal computed tomography (CT) from the University of Minnesota Medical Center between 2010 and 2018, with three specific tissues well-annotated (kidney, tumor, cyst). For AMOS dataset, we employ 200 multi-contrast abdominal CT with sixteen anatomies manually annotated for abdominal multi-organ segmentation. For MSD dataset, we employ in total of 585 abdominal contrast-enhanced CT scans for both pancreas and tumor (282) segmentation, and hepatic vessels and tumor (303) segmentation. More details of these four public datasets can be found in appendix A.2.

**Implementation Details** We specifically evaluate on direct supervised training scenario with all four datasets for volumetric segmentation. We perform five-fold cross-validations with 80% (train)/ 10% (validation)/ 10% (test) split to MSD and KiTS datasets, while single fold is performed with 80%

Table 1: Comparison of SOTA approaches on the three different testing datasets. (*: $p < 0.01$, with Paired Wilcoxon signed-rank test to all baseline networks)

| Methods | #Params | FLOPs | KiTS | | | | MSD | | | | | |
| | | | Kidney | Tumor | Cyst | Mean | Pancreas | Tumor | Mean | Hepatic | Tumor | Mean |
|---|---|---|---|---|---|---|---|---|---|---|---|---|
| 3D U-Net Çiçek et al. (2016) | 4.81M | 135.9G | 0.918 | 0.657 | 0.361 | 0.645 | 0.711 | 0.584 | 0.648 | 0.569 | 0.609 | 0.589 |
| SegResNet Myronenko (2018) | 1.18M | 15.6G | 0.935 | 0.713 | 0.401 | 0.683 | 0.740 | 0.613 | 0.677 | 0.620 | 0.656 | 0.638 |
| RAP-Net Lee et al. (2021) | 38.2M | 101.2G | 0.931 | 0.710 | 0.427 | 0.689 | 0.742 | 0.621 | 0.682 | 0.610 | 0.643 | 0.627 |
| nn-UNet Isensee et al. (2021) | 31.2M | 743.3G | 0.943 | 0.732 | 0.443 | 0.706 | 0.775 | 0.630 | 0.703 | 0.623 | 0.695 | 0.660 |
| TransBTS Wang et al. (2021) | 31.6M | 110.3G | 0.932 | 0.691 | 0.384 | 0.669 | 0.749 | 0.610 | 0.679 | 0.589 | 0.636 | 0.635 |
| UNETR Hatamizadeh et al. (2022b) | 92.8M | 82.5G | 0.921 | 0.669 | 0.354 | 0.648 | 0.735 | 0.598 | 0.667 | 0.567 | 0.612 | 0.635 |
| nnFormer Zhou et al. (2021) | 149.3M | 213.0G | 0.930 | 0.687 | 0.376 | 0.664 | 0.769 | 0.603 | 0.686 | 0.591 | 0.635 | 0.613 |
| SwinUNETR Hatamizadeh et al. (2022a) | 62.2M | 328.1G | 0.939 | 0.702 | 0.400 | 0.680 | 0.785 | 0.632 | 0.708 | 0.622 | 0.647 | 0.635 |
| 3D UX-Net (k=7) Lee et al. (2022) | 53.0M | 639.4G | 0.942 | 0.724 | 0.425 | 0.697 | 0.737 | 0.614 | 0.676 | 0.625 | 0.678 | 0.652 |
| UNesT-B Yu et al. (2023) | 87.2M | 258.4G | 0.943 | 0.746 | 0.451 | 0.710 | 0.778 | 0.601 | 0.690 | 0.611 | 0.645 | 0.640 |
| **DeformUX-Net (Ours)** | 55.8M | 635.8G | **0.948** | **0.763** | **0.450** | **0.720** | **0.790** | **0.643** | **0.717** | **0.637** | **0.705** | **0.671** |

(train)/ 10% (validation)/ 10% (test) split for AMOS dataset. The complete preprocessing and training details are available at the appendix A.1. Overall, we evaluate 3D DeformUX-Net performance by comparing with current volumetric transformer and CNN SOTA approaches for volumetric segmentation in fully-supervised setting. We leverage the Dice similarity coefficient as an evaluation metric to compare the overlapping regions between predictions and ground-truth labels. Furthermore, we performed ablation studies to investigate the best scenario of adapting the deformable convolution and the variability of substituting different linear layers for feature extraction.

## 6 RESULTS

### 6.1 EVALUATION ON ORGAN/VESSEL & TUMOR SEGMENTATION

We initiated our study by evaluating the adaptability across different organ/tissue scales through organ/vessel and tumor segmentation tasks. Table 1 provides a quantitative comparison against the leading state-of-the-art (SOTA) transformers and CNNs. In our analysis of the KiTS and MSD pancreas datasets, hierarchical transformers utilizing local attention mechanisms, such as SwinUNETR, outperformed the large kernel convolution mechanism employed by 3D UX-Net in MSD pancreas dataset, while the large kernel operation demonstrates better performance in KiTS. Given that the large kernel convolution statically summarizes features, the flexibility of local attention seems particularly advantageous when handling the sparsity within tumor regions. Our innovative convolution design subsequently improved performance metrics, pushing the mean organ dice from 0.680 to 0.720 in KiTS and 0.676 to 0.717 in MSD pancreas datasets. Notably, UNesT-L achieved performance metrics on par with our proposed block in the KiTS dataset, possibly due to its significant model capacity and unique local attention mechanism (experiments shown in supplementary material). Nevertheless, DeformUX-Net surpassed UNesT-B's performance in KiTS, boasting an enhancement of 1.41% in mean dice, and also outperformed UNesT-L on the MSD pancreas dataset, improving the Dice score from 0.701 to 0.717. As for vessel and tumor segmentation, where prior research has highlighted the vulnerabilities of large kernel convolution, 3D UX-Net still show significant strides over SwinUNETR. Our custom-designed block further elevated the performance, registering an increase of 4.84% in mean dice over both 3D UX-Net and UNesT-B. Moreover, qualitative visuals presented in Figure 3 clearly showcase our model's prowess in delineating organ boundaries and reducing the propensity for over-segmentation across adjacent organ regions.

### 6.2 EVALUATION ON MULTI-ORGAN SEGMENTATION

Apart from segmenting tissues across scales, Table 2 presents a quantitative comparison between the current SOTA transformers and CNNs for volumetric multi-organ segmentation. Employing our novel deformable convolution block design as the encoder backbone, the DeformUX-Net consistently outperforms its peers, showcasing a notable Dice score improvement ranging from 0.871 to 0.908. This enhancement is evident when compared to both SwinUNETR (which uses the Swin Transformer as its encoder backbone) and the 3D UX-Net (that relies on large kernel convolution for its encoder backbone). We also evaluate the effectiveness of DeformUX-Net against the most recent hierarchical transformer model, UNesT, known for its unique hierarchical block aggregation to capture local attention features. Remarkably, our deformable block design still outshined UNesT

Table 2: Evaluations on the AMOS testing split in different scenarios.(*: $p < 0.01$, with Paired Wilcoxon signed-rank test to all baseline networks)

| | | | | | | | Train From Scratch Scenario | | | | | | | | | |
|---|---|---|---|---|---|---|---|---|---|---|---|---|---|---|---|---|
| Methods | Spleen | R. Kid | L. Kid | Gall. | Eso. | Liver | Stom. | Aorta | IVC | Panc. | RAG | LAG | Duo. | Blad. | Pros. | Avg |
| nn-UNet | 0.951 | 0.919 | 0.930 | 0.845 | 0.797 | 0.975 | 0.863 | 0.941 | 0.898 | 0.813 | 0.730 | 0.677 | 0.772 | 0.797 | 0.815 | 0.850 |
| TransBTS | 0.930 | 0.921 | 0.909 | 0.798 | 0.722 | 0.966 | 0.801 | 0.900 | 0.820 | 0.702 | 0.641 | 0.550 | 0.684 | 0.730 | 0.679 | 0.783 |
| UNETR | 0.925 | 0.923 | 0.903 | 0.777 | 0.701 | 0.964 | 0.759 | 0.887 | 0.821 | 0.687 | 0.688 | 0.543 | 0.629 | 0.710 | 0.707 | 0.740 |
| nnFormer | 0.932 | 0.928 | 0.914 | 0.831 | 0.743 | 0.968 | 0.820 | 0.905 | 0.838 | 0.725 | 0.678 | 0.578 | 0.677 | 0.737 | 0.596 | 0.785 |
| SwinUNETR | 0.956 | 0.957 | 0.949 | 0.891 | 0.820 | 0.978 | 0.880 | 0.939 | 0.894 | 0.818 | 0.800 | 0.730 | 0.803 | 0.849 | 0.819 | 0.871 |
| 3D UX-Net | 0.966 | 0.959 | 0.951 | 0.903 | 0.833 | 0.980 | 0.910 | 0.950 | 0.913 | 0.830 | 0.805 | 0.756 | 0.846 | 0.897 | 0.863 | 0.890 |
| UNesT-B | 0.966 | 0.961 | 0.956 | 0.903 | 0.840 | 0.980 | 0.914 | 0.947 | 0.912 | 0.838 | 0.803 | 0.758 | 0.846 | 0.895 | 0.854 | 0.891 |
| DeformUX-Net (Ours) | **0.972** | **0.970** | **0.962** | **0.920** | **0.871** | **0.984** | **0.924** | **0.955** | **0.925** | **0.851** | **0.835** | **0.787** | **0.866** | **0.910** | **0.886** | **0.908\*** |

Table 3: Ablation studies of variable block and architecture designs on AMOS, KiTS, and MSD Pancreas datasets

| Methods | #Params (M) | AMOS | KiTS | Pancreas |
|---|---|---|---|---|
| | | | Mean Dice | |
| SwinUNETR | 62.2 | 0.871 | 0.680 | 0.708 |
| 3D UX-Net | 53.0 | 0.890 | 0.697 | 0.676 |
| UNesT-B | 87.2 | 0.894 | 0.710 | 0.690 |
| Use Standard Deformable Conv. | 55.5 | 0.878 | 0.701 | 0.682 |
| Use Depth-wise Deformable Conv. | 53.0 | 0.908 | 0.720 | 0.717 |
| Offset Kernel=$1 \times 1 \times 1$ | 52.5 | 0.908 | 0.720 | 0.717 |
| Offset Kernel=$3 \times 3 \times 3$ | 52.7 | 0.894 | 0.713 | 0.698 |
| x-y offset only Ying et al. (2020) | 54.0 | 0.878 | 0.701 | 0.689 |
| x-z offset only Ying et al. (2020) | 54.0 | 0.893 | 0.694 | 0.681 |
| y-z offset only Ying et al. (2020) | 54.0 | 0.883 | 0.712 | 0.697 |
| Tri-planar offset (Ours) | 55.8 | 0.908 | 0.720 | 0.717 |
| No MLP | 51.1 | 0.879 | 0.659 | 0.661 |
| Use MLP | 55.8 | 0.908 | 0.720 | 0.717 |
| Use Depth-wise Conv. Scaling Lee et al. (2022) | 53.0 | 0.889 | 0.684 | 0.679 |

across all organ evaluations, registering a 1.11% uptick in mean Dice score, while operating at a fifth of UNesT's model capacity. Further reinforcing our claims, Figure 3 visually underscores the segmentation quality improvements achieved by DeformUX-Net, illustrating its precision in preserving the morphology of organs and tissues in alignment with the ground-truth labels.

## 6.3 ABLATION ANALYSIS

After assessing the foundational performance of DeformUX-Net, we delved deeper to understand the contributions of its individual components—both within our novel deformable operation and the overarching architecture. We sought to pinpoint how these components synergize and contribute to the observed enhancements in performance. To this end, we employed the AMOS, KiTS, and MSD pancreas datasets for comprehensive ablation studies targeting specific modules. All ablation studies are conducted with kernel size $3 \times 3 \times 3$.

**Comparing with Different Convolution Mechanisms:** We examined both standard deformable convolution and depth-wise deformable convolution for feature extraction. To ensure a fair comparison between the experiments, we computed the deformable offset using a kernel size of $1 \times 1 \times 1$. Utilizing standard deformable convolution led to a notable decrease in performance across all datasets, while simultaneously increasing the model parameters. Conversely, by employing deformable convolution in a depth-wise manner, we found that independently extracting features across each channel was the most effective approach for subsequent segmentation tasks.

**Variation of Kernel Sizes for Offset Computation:** We observed a notable decrease when varying the method of offset computation. Using a MLP for offset calculation led to a significant performance improvement compared to the traditional method, which computes offsets with a kernel size of $3 \times 3 \times 3$. Intriguingly, by drawing inspiration from the parallel branch design in structural

re-parameterization, we managed to reduce the model parameters while maintaining performance across all datasets.

**Variability of Offset Directions:** Our study, drawing inspiration from Ying et al. (2020), delves into how the calculation of offsets in different directions impacts performance. Notably, the results hinge dramatically on the specific offset direction. For instance, in vessel segmentation, offsets determined for the x-z plane considerably outperform those in the other two planes. Conversely, offsets in the x-y plane are found to be more advantageous for multi-organ segmentation tasks. Given these findings, tri-planar offsets emerge as the most effective strategy for volumetric segmentation across various organ and tissue scenarios.

**Linear Scaling with MLP:** In addition to the primary deformable convolution operations, we examined the interplay between the convolution mechanism and various linear scaling operators. Adapting an MLP for linear scaling yielded substantial performance improvements across all datasets. In contrast, the gains from the depth-wise scaling, as introduced in 3D UX-Net, were more modest. To further amplify the spatial aggregation among adjacent features, the MLP proved to be an efficient and impactful choice.

## 7 DISCUSSION

In this work, we delve into the nuanced intricacies and associated trade-offs between convolution and self-attention mechanisms. Our goal is to introduce a block-wise design that addresses the shortcomings of prevailing SOTA operators, particularly large kernel convolution and global-local self-attention. Taking cues from the 3D UX-Net, we integrate our design within a "U-Net"-like architecture using skip connections tailored for volumetric segmentation. Beyond merely broadening the ERF for generic performance boosts, we discern that achieving excellence in dense prediction tasks hinges on two pivotal factors: **1) Establishing an optimal ERF for feature computation**, and **2) Recognizing and adapting to the spatial significance among adjacent features**. Simply enlarging convolution kernel sizes doesn't invariably enhance segmentation; performance might plateau or even regress without targeted guidance during optimization phases. Concurrently, the hierarchical transformer revitalizes local self-attention by computing features within specific sliding windows (e.g., $7 \times 7 \times 7$), albeit at the cost of some long-range dependency. Our insights from Tables 1 & 2 underscore that while large kernel convolution excels in multi-organ segmentation, global-local self-attention outperforms particularly in tumor segmentation. Traditional convolutions, being static, don't cater to spatial importance variations between neighboring regions in the way that self-attention does. This aspect of self-attention is especially potent in addressing tumor region sparsity. To address these challenges, we position deformable convolution as a promising alternative, distinguished by its inherent properties. Our adapted deformable mechanism emphasizes long-range dependencies even with smaller kernels, and we've expanded this approach in a depth-wise setting to emulate self-attention behavior. Drawing inspiration from structural re-parameterization's parallel branch design, we employ MLPs to compute deformable offsets. This facilitates discerning the importance and inter-relationships between neighboring voxels, which, when incorporated into feature computation, bolsters volumetric performance. Consequently, our novel design offers marked improvements over traditional designs employing standard deformable convolution.

## 8 CONCLUSION

We introduce DeformUX-Net, the first volumetric network tackling the trade-offs across CNNs to ViTs for medical image segmentation. We re-design the encoder blocks with depth-wise deformable convolution and projections to simulate all distinctive advantages from self-attention mechanism to large kernel convolution operations. Furthermore, we adapt tri-planar offset computation with a parallel branch design in the encoder block to further enhance the long-range dependency and adaptive spatial aggregation with small kernels. DeformUX-Net outperforms current transformer SOTAs with fewer model parameters using four challenging public datasets in direct supervised training scenarios.

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

# A  APPENDIX

## A.1  DATA PREPROCESSING & MODEL TRAINING

We apply hierarchical steps for data preprocessing: 1) intensity clipping is applied to further enhance the contrast of soft tissue (AMOS, KiTS, MSD Pancreas:{min:-175, max:250}; MSD Hepatic Vessel:{min:0, max:230}). 2) Intensity normalization is performed after clipping for each volume and use min-max normalization: $(X - X_1)/(X_{99} - X_1)$ to normalize the intensity value between 0 and 1, where $X_p$ denote as the $p_{th}$ percentile of intensity in $X$. We then randomly crop sub-volumes with size $96 \times 96 \times 96$ at the foreground and perform data augmentations, including rotations, intensity shifting, and scaling (scaling factor: 0.1). All training processes with 3D DeformUX-Net are optimized with an AdamW optimizer. We trained all models for 40000 steps using a learning rate of 0.0001 on an NVIDIA-Quadro RTX A6000 across all datasets. One epoch takes approximately about 9 minute for KiTS, 5 minutes for MSD Pancreas, 12 minutes for MSD hepatic vessels and 7 minutes for AMOS, respectively. We further summarize all the training parameters with Table 4.

Table 4: Hyperparameters for direction training scenario on four public datasets

| Hyperparameters | Direct Training |
|---|---|
| Encoder Stage | 4 |
| Layer-wise Channel | $48, 96, 192, 384$ |
| Hidden Dimensions | 768 |
| Patch Size | $96 \times 96 \times 96$ |
| No. of Sub-volumes Cropped | 2 |
| Training Steps | 40000 |
| Batch Size | 2 |
| AdamW $\epsilon$ | $1e-8$ |
| AdamW $\beta$ | $0.9, 0.999)$ |
| Peak Learning Rate | $1e-4$ |
| Learning Rate Scheduler | ReduceLROnPlateau |
| Factor & Patience | 0.9, 10 |
| Dropout | X |
| Weight Decay | 0.08 |
| Data Augmentation | Intensity Shift, Rotation, Scaling |
| Cropped Foreground | ✓ |
| Intensity Offset | 0.1 |
| Rotation Degree | $-30°$ to $+30°$ |
| Scaling Factor | x: 0.1, y: 0.1, z: 0.1 |

## A.2 Public Datasets Details

Table 5: Complete overview of Four public datasets

| Challenge | AMOS | MSD Pancreas | MSD Hepatic Vessels | KiTS |
|---|---|---|---|---|
| Imaging Modality | Multi-Contrast CT | Multi-Contrast CT | Venous CT | Arterial CT |
| Anatomical Region | Abdomen | Pancreas | Liver | Kidney |
| Sample Size | 361 | 282 | 303 | 300 |
| Anatomical Label | Spleen, Left & Right Kidney, Gall Bladder, Esophagus, Liver, Stomach, Aorta, Inferior Vena Cava (IVC) Pancreas, Left & Right Adrenal Gland (AG), Duodenum | Pancreas, Tumor | Hepatic Vessels, Tumor | Kidney, Tumor |
| Data Splits | 1-Fold (Internal) Train: 160 / Validation: 20 / Test: 20 | Train: 225 / Validation: 27 / Testing: 30 | 5-Fold Cross-Validation Training: 242, Validation: 30 / Testing: 31 | Training: 240, Validation: 30 / Testing: 30 |
| 5-Fold Ensembling | N/A | X | ✓ | X |

## A.3 Further Discussions Comparing with UNesT-L and RepUX-Net

Table 6: Evaluations on the AMOS testing split with UNesT-L and RepUX-Net.(*: $p < 0.01$, with Paired Wilcoxon signed-rank test to all baseline networks)

| | | | | | | | | | | | | | | | | Train From Scratch Scenario |
|---|---|---|---|---|---|---|---|---|---|---|---|---|---|---|---|---|
| Methods | Spleen | R. Kid | L. Kid | Gall. | Eso. | Liver | Stom. | Aorta | IVC | Panc. | RAG | LAG | Duo. | Blad. | Pros. | Avg |
| SwinUNETR | 0.956 | 0.957 | 0.949 | 0.891 | 0.820 | 0.978 | 0.880 | 0.939 | 0.894 | 0.818 | 0.800 | 0.730 | 0.803 | 0.849 | 0.819 | 0.871 |
| 3D UX-Net | 0.966 | 0.959 | 0.951 | 0.903 | 0.833 | 0.980 | 0.910 | 0.950 | 0.913 | 0.830 | 0.805 | 0.756 | 0.846 | 0.897 | 0.863 | 0.890 |
| UNesT-B | 0.966 | 0.961 | 0.956 | 0.903 | 0.840 | 0.980 | 0.914 | 0.947 | 0.912 | 0.838 | 0.803 | 0.758 | 0.846 | 0.895 | 0.854 | 0.891 |
| UNesT-L | 0.971 | 0.967 | 0.958 | 0.904 | 0.854 | 0.981 | 0.919 | 0.951 | 0.915 | 0.842 | 0.822 | 0.780 | 0.842 | 0.890 | 0.874 | 0.898 |
| RepUX-Net | 0.972 | 0.963 | **0.964** | 0.911 | 0.861 | 0.982 | 0.921 | **0.956** | 0.924 | 0.837 | 0.818 | 0.777 | 0.831 | **0.916** | 0.879 | 0.902 |
| DeformUX-Net (Ours) | **0.972** | **0.970** | 0.962 | **0.920** | **0.871** | **0.984** | **0.924** | 0.955 | **0.925** | **0.851** | **0.835** | **0.787** | **0.866** | 0.910 | **0.886** | **0.908*** |

Table 7: Comparison of UNesT-L and RepUX-Net on the three different testing datasets. (*: $p < 0.01$, with Paired Wilcoxon signed-rank test to all baseline networks)

| | | | KiTS | | | | MSD | | | | | |
|---|---|---|---|---|---|---|---|---|---|---|---|---|
| Methods | #Params | FLOPs | Kidney | Tumor | Cyst | Mean | Pancreas | Tumor | Mean | Hepatic | Tumor | Mean |
| SwinUNETR Hatamizadeh et al. (2022a) | 62.2M | 328.1G | 0.939 | 0.702 | 0.400 | 0.680 | 0.785 | 0.632 | 0.708 | 0.622 | 0.647 | 0.635 |
| 3D UX-Net (k=7) Lee et al. (2022) | 53.0M | 639.4G | 0.942 | 0.724 | 0.425 | 0.697 | 0.737 | 0.614 | 0.676 | 0.625 | 0.678 | 0.652 |
| UNesT-B Yu et al. (2023) | 87.2M | 258.4G | 0.943 | 0.746 | 0.451 | 0.710 | 0.778 | 0.601 | 0.690 | 0.611 | 0.645 | 0.640 |
| UNesT-L Yu et al. (2023) | 279.5M | 597.8G | 0.948 | **0.774** | **0.479** | **0.734** | 0.784 | 0.630 | 0.707 | 0.620 | 0.693 | 0.657 |
| **DeformUX-Net (Ours)** | 55.8M | 635.8G | **0.948** | 0.763 | 0.450 | 0.720 | **0.790** | **0.643** | **0.717** | **0.637** | **0.705** | **0.671** |

In Tables 6 and 7, we evaluate our proposed DeformUX-Net against two cutting-edge models: the UNesT, a transformer with a larger model capacity, and RepUX-Net, which employs the substantial 3D kernel size of $21 \times 21 \times 21$ for multi-organ segmentation. While UNesT-L surpasses both 3D UX-Net and SwinUNETR in several tasks, it underperforms in pancreas and tumor segmentation. This performance boost in UNesT-L could primarily be attributed to its unique ability to aggregate local attention and its vast model capacity, at 279.5M parameters. However, DeformUX-Net still excels in pancreas tumor and hepatic vessel tumor segmentation, improving mean organ dice scores by 1.41% and 2.13%, respectively, despite utilizing five times fewer model parameters. For kidney tumor segmentation, DeformUX-Net shows performance on par with UNesT-L. In multi-organ segmentation contexts, DeformUX-Net decisively outshines UNesT-L, enhancing the Dice score by 1.11% across all organs. RepUX-Net, meanwhile, exhibits performance closely aligned with UNesT-L, achieving a mean organ Dice of 0.902. Notably, the surge in segmentation performance from RepUX-Net arises from its innovative training strategy, which specifically addresses the challenges of local learning when employing a vast kernel size. This strategy elucidates the balance between global and local range dependencies essential for volumetric segmentation. Unlike RepUX-Net, our newly crafted block does not rely on any prior knowledge during network optimization, yet it consistently surpasses RepUX-Net with a mean organ Dice of 0.908 across most organs.

## A.4 Limitations of DeformUX-Net

While DeformUX-Net exhibits significant effectiveness across various segmentation scenarios when compared to diverse state-of-the-art approaches, it's not without limitations—both inherent to the deformable convolution operation and our specific block design. In our research, we limited our exploration to the efficacy of deformable convolution using a kernel size of $3 \times 3 \times 3$. We hypothesize that larger kernel sizes, such as $7 \times 7 \times 7$ could further improve segmentation performance. However,

enlarging the kernel brings challenges: both model parameters and computation scale exponentially when deformable offsets are integrated, and current 3D deformable convolution implementations in PyTorch are not as mature as their standard and depth-wise counterparts. Our work here is primarily empirical, aiming to showcase how deformable convolution can be robustly adapted for medical image segmentation. A potential future direction could delve into the engineering aspects of efficient deformable convolution, inviting further investigation into convolutional variants.

Beyond the operator computation, we also identified challenges in the computation of deformable offsets. Drawing from structural re-parameterization, our approach computes the offset using a parallel branch of the MLP module. This design, however, compromises efficiency during both training and testing phases. As kernel sizes for deformable convolution increase, the parallel branch kernel size must also scale concurrently, as inspired by Ding et al. (2022). A potential solution might lie in gradient re-parameterization, which could improve efficiency in both training and testing. This method would allow for the adaptation of deformable convolution without the need for parallel branch offset computation, as posited by Ding et al..

