# OpenReview forum: "DeformUX-Net: Exploring a 3D Foundation Backbone for Medical Image Segmentation with Depthwise Deformable Convolution"
_ICLR.cc/2024/Conference — ICLR 2024 Conference Withdrawn Submission_

### Official Review · Reviewer_HVYF · 2023-10-29

**Soundness:** 1 poor
**Presentation:** 3 good
**Contribution:** 2 fair
**Rating:** 3
**Confidence:** 5

**Summary:**

The paper introduces a 3D convolutional neural network model called DeformUX-Net for medical image segmentation. The model combines the advantages of Vision Transformers (ViTs) and large kernel depthwise convolution to address the limitations of existing core operators. The authors propose the use of deformable convolution in a depth-wise setting to achieve long-range dependency and adaptive spatial aggregation with computational efficiency. The model outperforms several existing models on four public medical image segmentation datasets, AMOS, MSD Pancreas, MSD Hepatic Vessels, and KiTS.

**Strengths:**

1. The study topic on exploring the 3D foundation backbone for medical image segmentation is significant, which will be benefit to contouring during radiotherapy planning.
2. The comparative results in Tables 1 and Table 2 show that the proposed DeformUX-Net can perform better than several CNN-based and Tansformer-based competitors on four public datasets.
3. Good organization. Most figures and tables are good and easy to understand.

**Weaknesses:**

1. Novelty Concerns: The proposed method mirrors the CVPR 2023's Internimage model. It seems to apply Internimage to 3D medical imaging without introducing fresh insights.
2. Deformable Convolution Review: Deformable convolution has been studied for years across domains. This paper misses a thorough discussion comparing its designs in computer vision and medical imaging.
3. Experimental Result Issues: The results lack conviction in their superior performance. The authors didn't employ official test sets, and the reproduced results, like those for nnUnet on KiTS (Kidney: 0.943 Dice, Tumor: 0.732 Dice), are notably lower than the original paper's metrics (Kidney: 0.9629 Dice, Tumor: 0.8420 Dice).

**Questions:**

1. Lack of novelty and new insights: In research, novelty is as critical as effectiveness. The method proposed closely aligns with the Internimage model from CVPR 2023. In the reviewer’s opinion, this work appears to be an application of Internimage to 3D medical image segmentation without delivering new perspectives to the community. Are there any advances in application or theory that set this work apart from Internimage?

2. Deformable Convolution Review: The deformable convolution design has been researched for years and has applications across various domains. The paper lacks a systematic discussion and comparison of the existing deformable convolution designs in both computer vision and medical imaging.

3. Inconvincing experimental Results: The experimental results are not entirely convincing regarding its state-of-the-art performance. Firstly, the authors haven't used official test sets for comparisons. To the reviewer's knowledge, online tests for KiTS and AMOS are available, and they recommend using these official sets for a fair evaluation. Moreover, the reproduced results, such as for nnUnet's performance on the KiTS dataset (Kidney: 0.943 Dice, Tumor: 0.732 Dice), are significantly lower than those reported in the official paper (Kidney: 0.9629 Dice, Tumor: 0.8420 Dice). The reviewer suggests a rigorous evaluation on official test sets to verify the findings.

4. Computational Complexity: The computational complexity of deformable depthwise convolution, especially in inference, compared to standard deformable convolution, remains unclear due to unfair comparisons in Table 1. In Table 3, only parameter quantities are compared. It's advisable to include FLOPS for an objective comparison.

5. Offset Kernel Performance: In Table 3, the performance of Offset Kernel 3 × 3 × 3 lags behind that of Offset Kernel 1 × 1 × 1. There's a lack of in-depth analysis in Section 6.3 (ABLATION ANALYSIS) to elucidate why a larger Offset Kernel adversely affects performance.

6. Comprehensive Metrics: While the paper might have touched upon some metrics, medical imaging segmentation often requires a multi-faceted evaluation. The Hausdorff Distance (HD), for instance, provides insights into the worst-case distances between two point sets - a vital metric in understanding segmentation accuracy. The absence of such metrics leaves the evaluation incomplete and raises questions about potential performance blind spots.

---

### Official Review · Reviewer_QwHv · 2023-10-30

**Soundness:** 3 good
**Presentation:** 3 good
**Contribution:** 3 good
**Rating:** 5
**Confidence:** 4

**Summary:**

This work introduces DeformUX-Net for volumetric medical image segmentation tasks. It leverages deformable convolution in a depth-wise setting with tri-planar offsets computed in a parallel branch design to adapt long-range dependency and adaptive spatial aggregation with computational efficiency. This paper uses four challenging public datasets to evaluate 3D DeformUX-Net in direct training scenarios with volumetric multi-organ/tissue segmentation across scales.

**Strengths:**

1. The proposed model DeformUX-Net the trade-offs between ViTs and large kernel convolution for 3D volumetric segmentation.

2. The proposed DeformUX-Net outperforms other SOTA models across all datasets evaluated, regardless of organ scale.

**Weaknesses:**

1. Novelty concern: DeformUX-Net builds upon the 3D UX-Net [r1] architecture and improves upon it by introducing a deformable operation that addresses the trade-offs inherent in global-local self-attention mechanisms and convolutions for volumetric dense predictions. Compared with [r1], the two papers are very similar in structure, research objectives, and writing style, and DeformUX-Net changes DWC to DDC.

2. This manuscript does not provide a detailed analysis of the computational requirements or training/test time for DeformUX-Net, which may be important considerations for their contribution.

[r1] Lee H H, Bao S, Huo Y, et al. 3D UX-Net: A Large Kernel Volumetric ConvNet Modernizing Hierarchical Transformer for Medical Image Segmentation[C]//The Eleventh International Conference on Learning Representations. 2022.

**Questions:**

1. Compared to Table 3 in this manuscript and Table 3 in [r1], it seems that DeformUX-Net cannot use a larger kernel size like  5^3, 7^3.

2. The performances of different methods (e.g., 3D UX-Net, nn-UNet, SWINUNETR) on AMOS dataset are inconsistent in Table 2 and [r1] (Table 2).

3. Did the author tested the model performance on Feta2021 and FLARE2021?

---

### Official Review · Reviewer_NgCv · 2023-10-31

**Soundness:** 3 good
**Presentation:** 4 excellent
**Contribution:** 3 good
**Rating:** 5
**Confidence:** 4

**Summary:**

This paper proposes DeformUX-Net, a volumetric deformable CNN segmentation model that attempts to balance and optimize three major trade-offs from current ViT and CNN research: global-local range dependency, adaptive spatial aggregation (deformable kernel via offsets), and computation efficiency. The major contribution is the block design (Figure 1). Experiments on five public segmentation datasets were performed, against SOTA ViT and CNN methods.

**Strengths:**

-	Sound conceptual justification of proposed block design, with evidence drawn from the progression of previous CNN and ViTs
-	Consistently superior results on all tasks attempted (if not against UNesT-L & RepUX-Net)

**Weaknesses:**

-	Effect of global-local range dependency tradeoff, one of the three major considerations mentioned, does not actually seem to be explored
-	Fairness of fixed training regime (with respect to number of steps and learning rate) as applied to all models compared against, is not clearly established

**Questions:**

1. A major concern is that all ablation studies were conducted with a kernel size of 3x3x3, with said kernel size described in Section 3 as controlling global-to-local range dependency; Section A.4 hypothesizes that larger kernels (e.g. 7x7x7) might improve segmentation at a tradeoff of model parameters and computation scale, but this tradeoff does not appear actually explored (with a smaller kernel if necessary)
2. Recent work on deformable 3D U-Nets might be discussed/compared against, e.g. Dong, Shunjie, et al. "DeU-Net 2.0: Enhanced deformable U-Net for 3D cardiac cine MRI segmentation." Medical Image Analysis 78 (2022): 102389.
3. The choice of the model hyperparameters presented in A.1, Table 4 and otherwise might be discussed, since it appears unlikely that the same set of hyperparameters would be optimal for all models. Were model hyperparameters independently optimized, and how?

---

### Official Review · Reviewer_jwaG · 2023-11-01

**Soundness:** 3 good
**Presentation:** 4 excellent
**Contribution:** 2 fair
**Rating:** 5
**Confidence:** 3

**Summary:**

This paper introduces the 3D DeformUX-Net, a volumetric CNN designed for medical image segmentation. It revisits the success of 3D ViTs and 3D CNNs and highlights the untapped potential of CNN-based models, especially those leveraging large kernel depthwise convolution. Central to this work's architecture is deformable convolution, aiming to enhance the advantages of several previous operations,  such as adaptive spatial aggregation, long-range dependency, and computational efficiency. In the reported experiments, the proposed method showcases its superiority by outperforming existing benchmarks.

**Strengths:**

1. Good framing. This paper starts by providing a comprehensive narrative that lays the foundation for their proposed method. Their exploration into the advantages and disadvantages of ViTs and large kernel CNNs, especially in the context of medical image segmentation, offers good insights.

2. Good results. The proposed 3D DeformUX-Net has achieved the best results in the tested experiments.

**Weaknesses:**

1. Limited novelty. The paper presents a compelling narrative, yet the novelty appears somewhat limited. Several other works have explored the concept of 3D deformable CNNs. Additionally, the triplane decomposition, although adapted for this context, has seen usage in various domains, including efficient attention mechanisms, 3D generative models, and neural radiance factorization. A more in-depth discussion contrasting this work with other 3D deformable CNNs and triplanes is needed.

 The paper would benefit from a side-by-side comparison with other 3D deformable CNN techniques. Demonstrating distinct advantages or highlighting unique approaches can help set 3D DeformUX-Net apart from existing works.

**Questions:**

Given the evident superior performance, it would be beneficial for the authors to further emphasize and clarify the novel aspects of their work, ensuring it stands out in the crowded space of medical image segmentation research.